# COP1 Deficiency in BRAF^V600E^ Melanomas Confers Resistance to Inhibitors of the MAPK Pathway

**DOI:** 10.3390/cells14130975

**Published:** 2025-06-25

**Authors:** Ada Ndoja, Christopher M. Rose, Eva Lin, Rohit Reja, Jelena Petrovic, Sarah Kummerfeld, Andrew Blair, Helen Rizos, Zora Modrusan, Scott Martin, Donald S. Kirkpatrick, Amy Heidersbach, Tao Sun, Benjamin Haley, Ozge Karayel, Kim Newton, Vishva M. Dixit

**Affiliations:** 1Department of Physiological Chemistry, Genentech, South San Francisco, CA 94080, USA; karayelo@gene.com (O.K.); knewton@gene.com (K.N.); dixit@gene.com (V.M.D.); 2Department of Neuroscience, Genentech, South San Francisco, CA 94080, USA; 3Department of Proteomics and Genomics Technologies, Genentech, South San Francisco, CA 94080, USA; rosec6@gene.com (C.M.R.); petrovicka83@yahoo.com (J.P.); modrusan@gene.com (Z.M.); dkirkpatrick2234@gmail.com (D.S.K.); 4Department of Molecular Oncology, Genentech, South San Francisco, CA 94080, USA; eqlin@gene.com (E.L.); martin.scott.everett@gmail.com (S.M.); 5Department of Computational Sciences, Genentech, South San Francisco, CA 94080, USA; rejar@gene.com (R.R.); s.kummerfeld@garvan.org.au (S.K.); a3blair5@gmail.com (A.B.); 6Melanoma Institute Australia, Wollstonecraft, NSW 2065, Australia; helen.rizos@mq.edu.au; 7Department of Biomedical Sciences, Faculty of Medicine, Health and Human Sciences, Macquarie University, Macquarie Park, NSW 2109, Australia; 8Department of Molecular Biology, Genentech, South San Francisco, CA 94080, USA; heidersa@gene.com (A.H.); sunt15@gene.com (T.S.); ben.haley@gmail.com (B.H.)

**Keywords:** COP1, DET1, BRAF, melanoma, vemurafenib

## Abstract

Aberrant activation of the mitogen-activated protein kinase (MAPK) cascade promotes oncogenic transcriptomes. Despite efforts to inhibit oncogenic kinases, such as BRAFV600E, tumor responses in patients can be heterogeneous and limited by drug resistance mechanisms. Here, we describe patient tumors that acquired COP1 or DET1 mutations after treatment with the BRAF^V600E^ inhibitor vemurafenib. COP1 and DET1 constitute the substrate adaptor of the E3 ubiquitin ligase CRL4^COP1/DET1^, which targets transcription factors, including ETV1, ETV4, and ETV5, for proteasomal degradation. MAPK-MEK-ERK signaling prevents CRL4^COP1/DET1^ from ubiquitinating ETV1, ETV4, and ETV5, but the mechanistic details are still being elucidated. We found that patient mutations in COP1 or DET1 inactivated CRL4^COP1/DET1^ in melanoma cells, stabilized ETV1, ETV4, and ETV5, and conferred resistance to inhibitors of the MAPK pathway. ETV5, in particular, enhanced cell survival and was found to promote the expression of the pro-survival gene BCL2A1. Indeed, the deletion of pro-survival BCL2A1 re-sensitized COP1 mutant cells to vemurafenib treatment. These observations indicate that the post-translational regulation of ETV5 by CRL4^COP1/DET1^ modulates transcriptional outputs in ERK-dependent cancers, and its inactivation contributes to therapeutic resistance.

## 1. Introduction

Activating mutations in the serine/threonine kinase BRAF cause aberrant mitogen-activated protein kinase (MAPK) signaling and are found in approximately 50% of malignant melanomas. Activating BRAF mutations also occur in a subset of lung (10%), thyroid (62%), and colorectal (20%) cancers [1]. In metastatic melanoma, 90% of BRAF mutations substitute valine 600 with glutamic acid (BRAFV^600E^), which favors the active conformation of the kinase without the need for activation signals transduced through RAS [2]. Vemurafenib is a small-molecule inhibitor of BRAF^V600E^ that is approved for the treatment of BRAF^V600E^ metastatic melanomas [3,4]. It is often combined with an inhibitor of MEK1/2, which are the kinases activated by BRAF [5]. Unfortunately, patients with BRAF^V600E^ mutant cancers rarely have durable responses to BRAF^V600E^ inhibition owing to intrinsic and acquired resistance mechanisms [6].

The RAS/RAF/MEK/ERK MAPK pathway couples extracellular signals to intracellular responses that include changes in gene transcription. In BRAF^V600E^ melanomas, a constitutively active MAPK pathway drives an ERK-dependent transcriptional output, and the inhibition of this output is correlated with a response to targeted therapies [7,8]. ERK and downstream kinases instigate gene expression by phosphorylating and activating transcription factors, including ETS domain-containing protein ELK1, proto-oncogene c-Myc, and AP-1 family members [6,9,10]. Whether other mechanisms couple ERK activity to transcriptional outputs in ERK-dependent cancers is unclear. There is interest in mutations that both promote transcription downstream of ERK and mediate melanoma resistance to the MAPK pathway because they could identify novel targets for pharmacologic intervention.

Cullin-4 (CUL4) functions as a key scaffold for forming ubiquitin chains, facilitating the transfer of ubiquitin through the RING protein RBX1 [11,12]. In CUL4 complexes, RBX1′s RING domain binds to E2 enzymes, transferring ubiquitin to target substrates via adapters recruited by DNA damage binding protein 1 (DDB1). DDB1 binds to the N-terminal region of CUL4 and recruits DCAFs, enhancing the specificity of CUL4–substrate interactions. Among DCAFs, De-etiolated-1 (DET1) is a highly conserved module, essential from plants to humans. DET1 forms the DDD complex with DDB1 and DDA1 [13,14,15], which uniquely interacts with E2 enzymes and recruits Constitutive Photomorphogenic 1 (COP1) into CRL4^COP1/DET1^ complexes [16]. Both DET1 and COP1 were initially identified in plants as regulators of light-mediated development [17]. In mammals, CRL4^COP1/DET1^ modulates various cellular processes, including tumorigenesis [18,19]. COP1 substrates include transcription factors such as c-JUN [16], ETV1, ETV4, ETV5 [19], ETS1, ETS2 [20], c/EBPα [21], and c/EBPβ [22]. The ubiquitination of these transcription factors by CRL4^COP1/DET1^ targets them for proteasomal degradation. Interestingly, ERK activation suppresses the ubiquitination of ETV5 by CRL4^COP1/DET1^, thereby reducing ETV5 protein turnover [23].

In this study, we identify patient tumors with de novo mutations in COP1 or DET1 after treatment with vemurafenib. These inactivating mutations stabilized ETV5 in a BRAF^V600E^ melanoma cell line and conferred resistance to MAPK pathway inhibition. RNA sequencing combined with chromatin analyses (ChIP and Hi-ChIP) revealed that ETV5 stabilization in COP1 mutant cells altered the expression of genes involved in cell identity, migration, proliferation, and survival. Notably, cells expressed more of the melanoma oncogene BCL2A1 (encoding BFL-1). Importantly, the deletion of BCL2A1 re-sensitized COP1 mutant BRAF^V600E^ cells to MAPK pathway inhibitors. Our observations indicate that the MAPK/COP1/ETV5 axis plays an important role in regulating transcriptional outputs in ERK-dependent cancers. Therefore, there may be therapeutic benefits to targeting both the MAPK pathway and BFL-1 in BRAF^V600E^ melanomas.

## 2. Materials and Methods

### 2.1. Cell Culture

We employed a Genentech-derived 293T cell line, a highly transfectable variant of the derivative of the 293 line, into which the temperature-sensitive gene for SV40 T-antigen was inserted. Cell lines 293T, A375, KPL4, and Cloudman S91 were cultured in Dulbecco’s Modified Eagle Medium (DMEM) supplemented with 10% fetal bovine serum, 2 mM glutamine, and 100 U/mL penicillin/streptomycin at 37 °C in a humidified incubator with 5% CO_2_.

The *COP1^KO.1^* and *COP1 ^KO.2^* A375 cell lines are clones derived from the same bulk A375 cells, which were edited using CRISPR/Cas9 technology. Two distinct guide RNAs were employed, CGCCGCTACCCGATACGGC and TGGTGGCGCCCGCCGTATC, targeting the first exon with respective specificity scores of 95 and 90. Similarly, the *ETV5^KO^* line was generated by editing *ETV5* via CRISPR/Cas9 technology in the A375 cell line, using guides ATAATCGCCCCAGTTACCAT and GTCGTCTTCTAGCCATGAGC, with respective specificity scores of 86 and 84. The K536E SNP was introduced into all three *COP1* alleles in A375 cells using CRISPR-nickase technology for targeted editing, which was subsequently validated via both Sanger and RNA sequencing.

### 2.2. Western Blots

Cells were lysed in 50 mM Tris HCl pH 7.4, 150 mM NaCl, 2 mM EDTA, 0.5% (*w*/*v*) Na-Deoxycholate, 0.1% (*w*/*v*) SDS, 1% (*v*/*v*) NP40, PhosSTOP phosphatase inhibitor (Roche, Basel, Switzerland), complete protease inhibitor cocktail (Roche, Basel, Switzerland), and 1 mM PMSF (Sigma-Aldrich, St. Louis, MO, USA). Protein quantitation was performed using the BCA Protein Assay kit (cat 23225, Pierce, Rockford, IL, USA). Antibodies recognized COP1 (28A4, Genentech, South San Francisco, CA, USA), ETV5 (4A6, Genentech, South San Francisco, CA, USA), ETV4 (1F7, Genentech, South San Francisco, CA, USA), ETV1 and ETV5 (13G11 Genentech, South San Francisco, CA, USA), ERK1/2 and phospho-ERK1/2 (9102, Cell Signaling, Danvers, MA, USA), beta-Actin (13E5, Cell Signaling, Danvers, MA, USA), FLAG (A8592 Clone M2, Sigma-Aldrich, St. Louis, MO, USA), HA (6E2, Cell Signaling, Danvers, MA, USA), BCL-XL (54H6, Cell Signaling, Danvers, MA, USA), BCL2 (D17C4, Cell Signaling, Danvers, MA, USA), MCL1 (D2W9E Cell Signaling, Danvers, MA, USA), BAX1 (D3R2M, Cell Signaling, Danvers, MA, USA), BAK (D4E4, Cell Signaling, Danvers, MA, USA), mouse BCL2A1 (6D6, [24]), human BFL-1 (6A7, [25]), and tubulin (DM1A, Cell Signaling, Danvers, MA, USA). Generally, an antibody dilution of 1:1000 or 1:2000 was employed.

### 2.3. Immunoprecipitations

Cells were lysed in 50 mM HEPES pH 7.2, 120 mM NaCl, 1 mM EDTA, 0.1% NP-40, PhosSTOP phosphatase inhibitor (Roche, Basel, Switzerland), and complete protease inhibitor cocktail (Roche, Basel, Switzerland). COP1 IPs were performed with anti-COP1 antibody (28A4, Genentech, South San Francisco, CA, USA) and magnetic protein A/G beads (cat. 88803, Pierce, Rockford, IL, USA). Immunoprecipitations with anti-FLAG M2 beads (Sigma-Aldrich, St. Louis, MO, USA) were eluted in 1 × LDS sample buffer (NP0007, Thermo Fisher Scientific, Waltham, MA USA).

### 2.4. Chemical Genomic Screen

A collection of 542 compounds listed in Appendix A was obtained from in-house synthesis or purchased from commercial vendors. Cells were maintained in RPMI-1640, 5% fetal bovine serum, and 2 mM glutamine in a humidified incubator maintained at 37 °C with 5% CO_2_. Cells were assessed with a Vi-CELL Cell Viability Analyzer (Beckman Coulter, Brea, CA, USA); a viability of at least 90% was required for screening. A Multidrop Combi Reagent Dispenser (Thermo Fisher Scientific, Waltham, MA USA) was used to plate 1000 cells into 384-well, black, clear-bottom plates (353962, Corning, NY, USA). Cells were treated with a 9-point dose titration of the chemical library using a Bravo Automated Liquid-Handling Platform (Agilent Technologies, Santa Clara, CA, USA). After 5 days, 25 µL CellTiter-Glo reagent was added using a MultiFlo Microplate Dispenser (BioTek Instruments, Winooski, VT, USA). Cell lysis was induced by mixing for 30 min on an orbital shaker. Plates were then incubated at room temperature for 10 min to stabilize the luminescent signal. Luminescence was read by a 2104 EnVision Multilabel Plate Reader (PerkinElmer, Waltham, MA, USA). The data was processed using Genedata Screener, Version 15, with a four-parameter Hill equation using compound dose–response data normalized to the median of 42 vehicle-treated wells on each plate. A Robust Fit strategy was also employed by Genedata Screener, which is based on Tukey’s biweight and is resistant to outlier data. The reported absolute IC50 is the dose at which cross-run estimated inhibition is 50% relative to DMSO control wells. In addition to absolute IC50, mean fitted viability across the nine tested doses (i.e., area under the viability curve) was also computed. Screen hits were defined as inhibitors associated with decreased sensitivity in COP1^KO^ or COP1^K539E^ cells compared to WT cells.

### 2.5. Cell Confluency Assays

Cells were seeded at a 10–20% density in DMEM supplemented with 10% fetal bovine serum, 2 mM glutamine, and 100 U/mL penicillin/streptomycin in 96-well plates (BA-04856, Incucyte^®^ Imagelock, Sartorius, Gottingen, Germany). Cells were treated with BRAF inhibitor vemurafenib, MEK inhibitor cobimetinib, or ERK inhibitor Vertex11e as indicated and were imaged with the IncuCyte S3 (Essen BioScience, Ann Arbor, MI, USA) at 10× magnification in the bright field channel every 2 h for 60–120 h. IncuCyte software was used to determine confluency over time (% confluency), and plots were generated using Prism (GraphPad v8).

### 2.6. Mass Spectrometry-Based Proteomics

Cell pellets were lysed in 8 M urea, 150 mM NaCl, 50 mM HEPES (pH 8.2), and a complete mini (EDTA-free) protease inhibitor (Roche, Basel, Switzerland). Protein concentration was determined by a BCA assay, and then disulfide bonds were reduced by incubation with 5 mM DTT (45 min, 37 °C). This was followed by alkylation of cysteine residues by 15 mM IAA (30 min, RT Dark), which was quenched by the addition of 5 mM DTT (15 min, RT Dark). Proteins were then extracted by chloroform/methanol precipitation and resuspended in a digestion buffer (8 M urea, 150 mM NaCl, 50 mM HEPES, pH 7.2). Initial protein digestion was performed by the addition of LysC (1:100 enzyme/substrate ratio), followed by incubation at 37 °C for 3 h. Samples were then diluted to 1.5 M urea with 50 mM HEPES (pH 7.2) before the addition of Trypsin (1:50 enzyme/substrate ratio) and incubated overnight at 37 °C. The next day, the resulting peptide mixtures were acidified and desalted via solid-phase extraction (SPE; SepPak, Waters, Milford, MA, USA). Each sample was then resuspended in 200 mM HEPES (pH 8.5) and labeled with tandem mass tags (TMT, Pierce, Rockford, IL, USA) according to the manufacturer’s instructions. Note, to enable analysis across multiplexed experiments, a bridge sample was created by mixing a small portion of each sample prior to TMT labeling. This sample was added at equal volume for each experiment within two “bridge channels”, which were used for normalization and quantification across multiplexed experiments. After 1 h of labeling, the reaction was quenched by the addition of 5% hydroxylamine and incubated at room temperature for 15 min. Labeled peptides were then mixed, acidified, and purified by SPE. The resulting peptide mixture was separated into 96 fractions by offline basic-reversed phase chromatography before being combined into 24 fraction pools, all of which were analyzed by LC-MS. For nLC-MS/MS analysis, peptides were separated using a Dionex UltiMate 3000 RSLCnano Proflow system (Thermo Fisher Scientific, Waltham, MA USA) with a gradient of 2% buffer A (98% H20, 2% ACN with 0.1% formic acid) to 30% buffer B (98% ACN, 2% H20, 0.1% formic acid) with a flow rate of 450 nL/min. Samples were separated over a 25 cm capillary column (100 µm I.D.) packed with Waters nanoAcquity M-Class BEH (1.7 µm) material (New Objective, Littleton, MA, USA). All samples were analyzed using an Orbitrap Fusion Lumos mass spectrometer (Thermo Fisher Scientific, Waltham, MA USA). For all analyses, the SPS-MS3 method was implemented for improved quantitative accuracy [26,27]. Intact peptides were surveyed in the Orbitrap (1 × 10^6^ AGC, 120,000 resolution), and the top 10 peptides were selected for fragmentation (CID, 30 NCE) and analyzed in the ion trap (2 × 10^4^ AGC). Quantitative MS3 scans selected the 8 most abundant fragment ions from the MS2 spectrum and fragmented them at high energy (HCD, 55 NCE, 2.5 × 10^5^ AGC) to produce reporter mass ions. 

### 2.7. Data Processing

Assignment of MS/MS spectra was performed using the MASCOT search algorithm to search against all entries for Mus musculus (house mouse) or Homo sapiens (human) in UniProt (downloaded June 2016). A search of all tryptic peptides (1 [mouse] or 2 [human] missed cleavages) was performed, and a precursor tolerance of 25 ppm [mouse] or 50 ppm [human] was used to limit the number of candidate peptides, while a 0.8 Da tolerance was used to match MS/MS data collected in the ion trap. Static modifications included TMT on the N-terminus of peptides and lysine residues (+229.16293) and cysteine alkylation (+57.0215), while variable modifications included methionine oxidation (+15.9949) for mouse and human data and TMT labeling of tyrosine (+229.1629) for mouse data. Peptide spectral matches were filtered to a 2% false discovery rate using a target decoy approach scored with a linear discrimination analysis algorithm before filtering to a 2% false discovery rate at the protein level as previously described [28]. Quantitative values were extracted and corrected for isotopic impurities using Mojave [29]. Additionally, quantitative events with a precursor purity < 0.5 (±0.25 Da) or sum intensity < 30,000 were discarded before quantitative values were normalized and converted to “relative abundance” values using custom scripts coded in R. For quantitative analysis, peptide spectral match level data was summed to the peptide level, and peptide level data was summed to the protein level. When appropriate, data were normalized to the bridge channel to enable analysis of data across multiplexed experiments. Data were further visualized in Spotfire (PerkinElmer, Waltham, MA, USA).

### 2.8. RNA-Sequencing

Total RNA was extracted using an RNaesy Mini Kit (cat74104, Qiagen, Redwood City, CA, USA) following the manufacturer’s protocol, and was quantified in a NanoDrop 8000 (Thermo Fisher Scientific, Waltham, MA USA). A total of 0.1 μg of total RNA was used for library preparation with a TruSeq Stranded Total RNA Library Prep Kit (Illumina, San Diego, CA, USA). The libraries were multiplexed and then sequenced on an Illumina HiSeq4000 (Illumina, San Diego, CA, USA) to generate 30 million single-end 50 bp reads. GNAP (version 2013-11-10; http://research-pub.gene.com/gmap/) [30] was used to align raw FASTQ files to the human reference genome (GRCh38) with parameters “-M 2 -n 10 -B 2 -i 1 -N 1 -w 200000 -E 1 --pairmax-rna=200000 --clip-overlap”. For mouse RNAseq data, the raw FASTQ files were mapped to mouse reference genome (GRCm38). Uniquely mapped reads were used for downstream analyses.

### 2.9. RNA-Seq Differential Gene Expression Analysis

The Limma R package was used to perform the differential gene expression analysis. Differences between samples were tested using a moderated version of the *t*-test. The resulting p-values were adjusted for multiple comparisons by controlling the false discovery rate (FDR) using the Benjamini–Hochberg method. Genes were considered differentially expressed if the log2 fold change was >1 or <−1 and adjusted *p*-value < 0.05. R package pheatmap (Version 1.0.12) was used to create the heatmaps. Gene set scores were calculated using the GSVA R package.

### 2.10. ChIP-Sequencing

Cells were fixed with 1% formaldehyde for 15 min and quenched with 0.125 M glycine. Chromatin was isolated by the addition of lysis buffer, followed by disruption with a Dounce homogenizer. Lysates were sonicated and the DNA sheared to an average length of 300–500 bp. Genomic DNA (input) was prepared by treating aliquots of chromatin with RNase, proteinase K, and heat for de-crosslinking, followed by ethanol precipitation. Pellets were resuspended, and the resulting DNA was quantified on a NanoDrop spectrophotometer. Extrapolation to the original chromatin volume allowed for the quantitation of the total chromatin yield. An aliquot of chromatin (25 µg) was precleared with protein Agarose beads (Invitrogen, Thermo Fisher Scientific, Waltham, MA, USA). Genomic DNA regions of interest were isolated using 4 µL of antibody against ETV5 (ab102010, Abcam, Waltham, MA, USA) and H3K27Ac (39133, Active Motif, Carlsbad, CA, USA) or 3 µL of H3K4me3 antibody (39159, Active Motif, Carlsbad, CA, USA). Complexes were washed, eluted from the beads with an SDS buffer, and subjected to RNase and proteinase K treatment. Crosslinks were reversed by incubation overnight at 65 °C, and ChIP DNA was purified by phenol-chloroform extraction and ethanol precipitation. Quantitative PCR (QPCR) reactions were carried out in triplicate on specific genomic regions using SYBR Green Supermix (Bio-Rad, Hercules, CA, USA). The resulting signals were normalized for primer efficiency by carrying out qPCR for each primer pair using input DNA. Illumina sequencing libraries were prepared from the ChIP and input DNAs by the standard consecutive enzymatic steps of end polishing, dA-addition, and adaptor ligation. Steps were performed on an automated system (Apollo 342, Wafergen Biosystems/Takara, San Jose, CA, USA). After a final PCR amplification step, the resulting DNA libraries were quantified and sequenced on Illumina’s NextSeq 500 (75 nt reads, single end). Reads were aligned to the human genome (hg38) using the BWA algorithm (default settings). Duplicate reads were removed, and only uniquely mapped reads (mapping quality ≥ 25) were used for further analysis.

### 2.11. Hi-ChIP

H3K27ac Hi-ChIP was performed using the Arima-HiC^+^ kit (A510008, Arima Genomics, Carlsbad, CA, USA) according to the manufacturer’s instructions. Briefly, cells were counted using the ViCell XR automated viability analyzer (Beckman Coulter, Brea, CA, USA) and crosslinked with 2% formaldehyde (F79-1, Thermo Fisher Scientific, Waltham, MA USA) for 10 min and subsequently quenched with 0.2 M glycine (AAJ1640736, Thermo Fisher Scientific, Waltham, MA, USA). Chromatin from 4 × 10^6^ cells (~12 μg) was digested, followed by biotin end filling ligation reactions and sonicated in 1 × Shearing Buffer D3 (truChIP chromatin shearing kit, 520154, Covaris, Woburn, MA, USA) on Covaris LE220 to obtain DNA fragments of ~600 bp. Sheared chromatin was incubated with the H3K27ac antibody (91193, Active Motif, Carlsbad, CA, USA) at 4 °C overnight. Chromatin–antibody complexes were captured with Protein-A magnetic beads (10002D, Invitrogen, Thermo Fisher Scientific, Waltham, MA, USA) and subsequently washed with Low Salt Wash Buffer, High Salt Wash Buffer, LiCl Wash Buffer, low EDTA TE Buffer, and eluted. De-crosslinked DNA was purified with SPRI beads (B23318, Beckman Coulter, Brea, CA, USA) and quantified using the Qubit dsDNA HS Assay Kit (Q32854, Invitrogen, Thermo Fisher Scientific, Waltham, MA, USA). Samples passing ChIP efficiency QC, Arima-QC1, were used for library construction with the Accel-NGS 2S Plus DNA Library Kit (21024, Swift Biosciences, Ann Arbor, MI, USA) and were further assessed with library complexity QC. Samples passing library complexity QC were subsequently indexed using the Swift Biosciences Accel-NGS 2S Set A indexing kit (26148, Swift Biosciences, Ann Arbor, MI, USA) and PCR amplified based on the manufacturer’s recommendations using HiFi HotStart master mix from the KAPA Library Amplification Kit (KK2620, Roche, Basel, Switzerland). Final Hi-ChIP libraries were quantified with TapeStation 4200 (Agilent Technologies, Santa Clara, CA, USA) and the KAPA Library Quantification Kit (KK4824, Roche, Basel, Switzerland) and paired-end (150 bp + 150 bp) sequenced on an Illumina NovaSeq instrument.

### 2.12. ChIP-Seq Differential Binding Analysis and Ontology Enrichment

MACS 2.1.0 was used to call peaks. The GREAT v4.0.4 web application was used to perform gene ontology (GO) and transcription start site (TSS) analysis on ETV5 peaks that were significantly enriched (log2 fold change > 0 and adjusted *p*-value < 0.05) in the COP1 SNP sample relative to COP1 WT. MEME suite was used on these ETV5 peaks to look for over-represented motifs. The WashU epigenome browser was used to visualize ChIP-seq data at different genomic loci.

## 3. Results and Discussion

### 3.1. Deletion or Inactivation of COP1 Renders Melanoma Cells Less Sensitive to Inhibitors of the MAPK Pathway

Given that ERK suppresses the activity of CRL4^COP1/DET1^ [23], we investigated whether COP1 and DET1 impact tumor responses to MAPK pathway inhibitors as reported by others [31]. Substrates of CRL4 ^COP1/DET1^, including c-JUN, ETV1, and ETV5, are degraded when ERK is inactive [13,23,31], but the critical transcriptional outputs of these transcription factors in melanoma are unclear. First, we interrogated the internal exome sequencing data, later published by Wongchenko et al. [32], for evidence of *COP1* mutations in patient melanomas after treatment with vemurafenib. One patient, who responded to treatment initially but then became unresponsive, had acquired a missense mutation *COP1^K539E^* (Figure 1A). Missense mutations in *COP1* were also found in 12 out of 50 patients whose melanomas never responded to vemurafenib. These mutations are largely localized to the C-terminal WD40 domain of COP1 that mediates substrate binding (Figure 1B). Another two non-responders had nonsense mutations in *COP1* that would eliminate the WD40 domain. These observations suggest that COP1 impairment might contribute to vemurafenib resistance.

To explore this possibility further, we deleted *COP1* from A375 melanoma cells bearing the BRAF^V600E^ mutation and then determined how the cells responded to a panel of 542 anti-cancer agents. The loss of *COP1* conferred significant resistance to compounds inhibiting BRAF^V600E^, MEK, or ERK (Figure 1C–F), supporting the notion that COP1 suppresses cell proliferation and survival when the MAPK pathway is inhibited. Sensitivity to other types of inhibitors, including taxol derivatives, mTOR inhibitors, HSP90 inhibitors, and PI3K inhibitors, was less affected by COP1 loss. Modifying the endogenous *COP1* locus to express COP1^K539E^ also protected A375 cells from MAPK pathway inhibitors (Figure 1G,H and Appendix A), implying that COP1^K539E^ is a loss-of-function mutant.

Vemurafenib and cobimetinib arrest cell proliferation by potently inhibiting BRAF and MEK1/2, respectively [2]. To confirm our initial screen results, we exposed WT, *COP1^K539E^*, and *COP1^KO^* A375 cells to different drug concentrations. *COP1^K539E^* or *COP1^KO^* cells consistently proliferated more than wild-type cells in the presence of vemurafenib or cobimetinib (Figure 1H and Appendix A). *COP1* knockdown using shRNAs has been reported to give similar results [31]. Collectively, these data indicate that COP1 is required for the normal sensitivity of A375 cells to MAPK pathway inhibition.

### 3.2. COP1 Substrates ETV1, ETV4, and ETV5 Are Rapidly Eliminated upon ERK Inhibition

We reasoned that increased CRL4^COP1/DET1^ activity as ERK signaling subsides must result in the degradation of a substrate(s) that promotes cell proliferation and/or survival. To identify the proteins that are degraded by COP1 in A375 cells treated with MAPK pathway inhibitors, we used an unbiased global proteomics strategy. Peptides that decreased in abundance the most within 30 min of treatment with the ERK inhibitor Vertex11e were either unique to ETV5 or shared between ETV1, ETV4, and ETV5 (Figure 2A–C). Western blotting confirmed that ERK inhibition reduced ETV1, ETV4, and ETV5 protein expression in WT but not *COP1^KO^* or *COP1^K539E^* cells (Figure 2D). These results indicate that CRL4^COP1/DET1^ is the only ubiquitin ligase to promote ETV1, ETV4, and ETV5 turnover in A375 cells upon ERK inhibition. This ERK, CRL4^COP1/DET1^, and ETV5 axis was not unique to A375 cells. ERK activity also inhibited COP1-dependent proteasomal degradation of ETV5 in human KPL-4 breast cancer cells and mouse Cloudman S91 melanoma cells (Appendix A).

To determine whether ETV1, ETV4, or ETV5 has the most impact on gene expression in the melanocyte lineage, we performed bulk RNA sequencing on primary melanocytes from WT, *Etv1^−/−^*, *Etv5^−/−^*, and *Etv1^−/−^ Etv4^−/−^ Etv5^−/−^* mice. *Etv5* mRNA was slightly more abundant than *Etv1* or *Etv4* mRNAs in WT melanocytes (Appendix A). In addition, the deletion of *Etv5* reduced the expression of a larger number of genes than the loss of *Etv1* (Appendix A). The combined deletion of *Etv1*, *Etv4*, and *Etv5* did not down-regulate more genes than the loss of *Etv5* alone. These data suggest that ETV5 is dominant over ETV1 and ETV4 in promoting gene expression in melanocytes. Therefore, in subsequent experiments, we focused on ETV5 dysregulation in COP1 mutant melanoma cells.

### 3.3. COP1^K539E^ Fails to Interact with the Transcription Factor ETV5

COP1 residue K539 sits in the substrate-binding WD40 domain [33] (Appendix A). Consistent with this residue being important for substrate binding, ETV5 in A375 cells coimmunoprecipitated with WT COP1 but not COP1^K539E^ (Figure 2E). This result suggests that ETV5 is not degraded in *COP1^K539E^* cells because it cannot interact with COP1^K539E^. To determine if altering the charge at nearby residue, K537, also compromised COP1, we examined ETV5 protein expression in 293T cells co-transfected with DET1 and either COP1^WT^, COP1^K537E^, or COP1^K539E^ (Figure 2F). In this setting, COP1^WT^ or COP1^K537E^ combined with DET1 to suppress ETV5 protein expression, whereas COP1^K539E^ did not. Therefore, the location of the charge swap in the WD40 domain determines whether COP1 can function or not. Other COP1 mutations in patient melanomas (*COP1^1−380^*, *COP1^1−384^*, *COP1^W517A^*, *COP1^W517L^*, and *COP1^R359Q^*) also compromised the interaction of COP1 with ETV5 (Figure 2G). Mutant COP1^K472E^, which cannot interact with the pseudokinase TRIB1 [34], also failed to bind to ETV5 (Figure 2G), consistent with TRIB1 and ETV5 binding to a common surface on COP1 [33]. Collectively, our results indicate that mutating select residues in the WD40 domain of COP1 prevents interactions with ETV5.

The crystal structure of the COP1 WD40 domain [33] shows K539 in proximity to W517, another residue that is important for interactions with ETV5 (Figure 2G). We hypothesized that salt bridges between K539 and residues E511 and E513 stabilize W517 (Appendix A). Therefore, the K539E mutation would disrupt the salt bridges and thereby perturb substrate binding. To test this notion, we introduced mutations E511K and/or E513K into COP1^K539E^ to favor salt bridge formation and then examined ETV5 binding. A small amount of ETV5 coimmunoprecipitated with COP1^E511K, K539E^, but swapping the charge at both glutamic acid residues in COP1^E511K, E513K, K539E^ restored ETV5 binding even more (Appendix A). This result indicates that electrostatic interactions within the COP1 WD40 domain are crucial for ETV5 binding.

### 3.4. Mutation of DET1 in a Vemurafenib-Resistant Melanoma

To further corroborate the role of CRL4^COP1/DET1^ in the response to inhibitors of the MAPK pathway, we also looked for evidence of *DET1* mutations in patient melanomas after treatment with vemurafenib. De novo missense mutation *DET1^S192F^* was identified in one metastatic melanoma [35]. *DET1^S192F^* is a loss-of-function mutation because the combination of DET1^S192F^ and COP1^WT^ failed to suppress ETV5 protein expression in 293T cells (Figure 2H).

Patient tumors were heterozygous for the *COP1^K539E^* or *DET1^S192F^* mutations, which suggested the mutant proteins might interfere with their WT counterparts. To explore this possibility, we introduced doxycycline-inducible Flag-tagged COP1 (^Flag^COP1WT) into the *COP1^K539E^* A375 cells. The expression of ^Flag^COP1^WT^ did not cause a marked reduction in ETV5 protein (Figure 2I). In addition, the cells were not greatly sensitized to the MEK inhibitor cobimetinib using a cell proliferation assay (Figure 2J). This result supports the notion that *COP1^K539E^* represents a dominant negative mutation.

### 3.5. ETV5 Protein Abundance Correlates with Resistance to MAPK Pathway Inhibition

To determine the importance of ETV5 to the drug resistance phenotype of *COP1^KO^* A375 cells, we deleted ETV5 from the cells. Western blotting of the *COP1^KO^ ETV5^KO^* cells confirmed ETV5 protein loss without any alteration in ETV4 or ETV1 protein expression (Appendix A). Remarkably, the loss of ETV5 was sufficient to re-sensitize the *COP1^KO^* cells to cobimetinib treatment (Appendix A). We reconstituted *COP1^KO^ ETV5^KO^* cells with ETV5^WT^ to confirm that their proliferation in the presence of cobimetinib was indeed due to ETV5 deficiency (Appendix A). We also reconstituted *ETV5^KO^* A375 cells with either ETV5^WT^ or ETV5^∆DEG^ (Appendix A), the latter an N-terminally truncated ETV5 lacking the two degron motifs that mediate interactions with COP1 [19]. We predicted that the expression of ETV5^∆DEG^ in *ETV5^KO^* cells expressing COP1^WT^ would mimic the expression of ETV5^WT^ in *Cop1^KO^ Etv5^KO^* cells because ETV5^∆DEG^ does not undergo COP1-dependent proteasomal degradation. Accordingly, ETV5^WT^ was only detected in reconstituted *Etv5^KO^* cells after treatment with the proteasome inhibitor MG-132, whereas ETV5^∆DEG^ was detected with or without MG-132 (Appendix A). Importantly, *Cop1^KO^ Etv5^KO^* cells reconstituted with ETV5^WT^ (or *Etv5^KO^* cells reconstituted with ETV5^∆DEG^) proliferated more than *Cop1^KO^ Etv5^KO^* cells (or *Etv5^KO^* cells reconstituted with ETV5^WT^) in the presence of the cobimetinib (Appendix A). We conclude from these experiments that the resistance of A375 cells to MAPK pathway inhibition stems, in large part, from aberrant expression of ETV5.

### 3.6. COP1^K539E^ Elicits a Gene Signature in Melanoma Cells That Is Associated with Resistance to MAPK Pathway Inhibitors

Others have suggested that ETV1, ETV4, and ETV5 modulate MAPK signaling by promoting the expression of negative regulators of the pathway, including dual-specificity phosphatase DUSP6 [31]. However, neither COP1 deficiency nor COP1^K539E^ altered the abundance of DUSP6 or DUSP4 (Appendix A). To gain insights into the consequences of ETV5 stabilization in *COP1^K539E^* cells, we looked for changes in gene expression. RNA sequencing revealed that 975 transcripts were increased more than 2-fold in *COP1^K539E^* cells compared with *COP1^WT^* cells (Figure 3A), while 1349 transcripts decreased more than 2-fold (Figure 3B). The down-regulated genes included *BACE2*, *TYRO3*, *MITF*, and *SOX10*, which are markers of terminally differentiated melanocytes. This result was intriguing because melanoma cells dedifferentiate under cellular stress, and this contributes to intrinsic and acquired resistance to MAPK pathway inhibitors.

We also compared the expression profiles of *COP1^WT^* and *COP1^K539E^* cells to melanoma subtypes defined using 53 human melanoma cell lines [36]. COP1^WT^ was enriched in a transitory/neural crest-like signature and acquired an undifferentiated signature when COP1 was mutated (Figure 3C). An undifferentiated subtype, characterized by enrichment of gene sets linked to cell migration and extracellular matrix organization, is known to confer resistance to targeted therapies, such as vemurafenib [37].

### 3.7. COP1^K539E^ Increases Expression of BCL2A1

*BCL2A1*, which encodes the pro-survival BCL-2 family member BFL-1, was among the top 50 differentially expressed genes in *COP1^K539E^* cells (Figure 3A). This gene is exclusively amplified in melanoma and has been shown to confer resistance to BRAF-directed therapy [38]. *BCL2A1* transcripts were 30-fold more abundant in *COP1^K539E^* cells than in *COP1^WT^* cells (Figure 3D), and this coincided with an increased expression of BFL-1 protein (Figure 3E). COP1 deficiency also increased *Bcl2a1* mRNA and protein expression in mouse Cloudman S91 cells (Figure 3E). These data suggest that vemurafenib and cobimetinib resistance owing to COP1 inactivation in melanoma cells may, in part, stem from an enhanced expression of BFL-1.

To determine whether *BCL2A1* and any of the other genes up- or down-regulated by COP1^K539E^ might be direct transcriptional targets of ETV5, we performed chromatin immunoprecipitation sequencing (ChIP-seq) of *COP1^WT^* and *COP1^K539E^* A375 cells using antibodies to ETV5, trimethylated lysine 4 on histone H3 (H3K4me3, a marker of active promoters), and acetylated lysine 27 on histone H3 (H3K27ac, a marker of active enhancers) (Figure 4A). The suppression of ETV5 protein expression in *COP1^WT^* cells upon ERK inhibition coincided with fewer sites in chromatin being bound by ETV5 (Figure 4B; middle bar). In contrast, ERK inhibition did not change the number of ETV5 binding sites in *COP1^K539E^* cells (Figure 4B; lower bar), reflecting ETV5 stability in these cells. Untreated *COP1^K539E^* cells, which express more ETV5 than untreated *COP1^WT^* cells, also exhibited more chromatin sites bound by ETV5 than *COP1^WT^* cells (Figure 4B; upper bar). This result suggests that the binding of ETV5 to additional sites in the genome may contribute to the resistance phenotype.

The majority of the ETV5 peaks in *COP1^K539E^* cells were 5–500 kb away from transcription start sites (TSSs), suggesting that ETV5 largely binds to enhancer regions (Figure 4C). De novo motif analysis of these distal ETV5 binding sites identified binding motifs for several transcription factors, including JUN/FOS, BATF, ETS1, and ZNF770 (Figure 4D). The genes closest to the distal ETV5 peaks were associated with the gene ontology terms “regulation of differentiation”, “regulation of extracellular matrix assembly”, and “regulation of tissue remodeling” (Figure 4E), but understanding the functional significance of these ETV5 binding sites will require further study.

When we analyzed *BCL2A1*, we found that the *BCL2A1* promoter exhibited more of the H3K4me3 activation mark in *COP1^K539E^* cells than in *COP1^WT^* cells (Figure 4F). This result is consistent with the former expressing more *BCL2A1* mRNA than the latter. *COP1^K539E^* cells also had much more ETV5 bound to the *BCL2A1* promoter region than *COP1^WT^* cells (Figure 4F), which might indicate the direct regulation of *BCL2A1* by ETV5. Given that ETV5 was mainly associated with distal enhancer regions, it might also regulate transcription through enhancer repositioning and promoter–enhancer interactions. Hi-ChIP analysis indicated that *COP1^K539E^* cells had altered contacts between active enhancers, marked by H3K27ac, and active promoter regions, marked by H3K4me3, for several genes, including *BCL2A1* and *MCL1* (Figure 4G and Appendix A). However, the significance of these changes and their dependence on ETV5 will require further study.

### 3.8. ETV5-Driven Expression of BCL2A1 Confers Resistance to MAPK Pathway Inhibitors

To address whether BFL-1 protected *COP1^K539E^* A375 cells from MAPK pathway inhibitors, we generated *COP1^K539E^ BCL2A1^KO^* cells (Figure 5A). These cells proliferated, as well as *COP1^K539E^* and *COP1^WT^* cells, in the absence of drug treatment (Figure 5B), indicating that BFL-1 was dispensable for the survival of the unstressed cells. However, *COP1^K539E^ BCL2A1^KO^* cells did not proliferate as well as *COP1^K539E^* cells in the presence of either vemurafenib or cobimetinib (Figure 5B). Thus, BFL-1 expression modulates the sensitivity of A375 cells to MAPK pathway inhibitors.

To determine the contribution of BFL-1 to the resistance phenotype of *COP1^K539E^* cells relative to the other pro-survival BCL2 family members, we screened inhibitors of the different BCL2 pro-survival proteins for their ability to re-sensitize *COP1^K539E^* cells to cobimetinib (Figure 5C). Individually, the BCL2 (GDC-0199), BCL-XL (A-1331852), and MCL1 (AMG-176) inhibitors partially sensitized the *COP1^K539E^* cells, but the best sensitization was achieved with either all three inhibitors combined or when inhibitors of BCL2 and MCL1 were combined with BFL-1 deficiency. Indeed, BFL-1 was recently shown to mediate resistance to combined MCL1 and BCL-XL inhibition in some melanoma cell lines, consistent with all three pro-survival proteins countering the effects of MAPK pathway inhibition [39].

## 4. Discussion and Conclusions

Here, we report that the mutational inactivation of the Cullin-4 E3 ubiquitin ligase substrate adaptor COP1 serves as a mechanism of resistance to MAPK-targeted therapy in melanoma patients. Specifically, an internal exome sequencing dataset later published in [32] revealed a novel *COP1* mutation, p.K539E, in a patient’s melanoma that became unresponsive to vemurafenib after an initial response. This mutation occurs in the C-terminal WD40 domain of COP1, which is responsible for substrate binding and, consequently, disrupts COP1’s interaction with its substrates. As a result, ETS family transcription factors, known substrates of CRL4^COP1/DET1^, are stabilized in *COP1^K539E^* melanoma cells because they are no longer ubiquitinated and targeted for proteasomal degradation. We demonstrated that ETV5 particularly contributes to drug resistance and cell survival by promoting the expression of pro-survival genes, including *BCL2A1*. The binding of ETV5 to the *BCL2A1* promoter suggests that BCL2A1 might be targeted by ETV5 directly. Importantly, the deletion of *BCL2A1* re-sensitized COP1-deficient *BRAF^V600E^* cells to MAPK pathway inhibition. Therefore, combining therapies that target the MAPK pathway as well as BFL-1, the protein encoded by *BCL2A1*, may enhance tumor regression.

Prior research [31] has demonstrated that deficiencies in COP1 or DET1 lead to elevated levels of ETS transcription factors. This elevation effectively decouples MAPK signaling from transcriptional regulation, thereby contributing to resistance against MAPK inhibitors. However, their proposed mechanisms underlying this resistance differ. Their research suggests that COP1 deficiency not only causes the accumulation of ETS factors but also increases the levels of MAPK negative feedback regulators, such as the DUSP6 phosphatase, resulting in reduced ERK phosphorylation. In contrast, our findings indicate that COP1 deficiency does not affect the expression of DUSP6 and DUSP4. Furthermore, their study provides a broader perspective across various cancer types, while our research is specifically focused on melanoma patients, with potential implications for developing combination therapies that target both the MAPK pathway and specific transcriptional regulators, like BFL-1.

Supporting CRL4^COP1/DET1^’s role in the response to inhibitors of the MAPK pathway, a de novo missense *DET1* mutation, p.S192F, was also discovered in vemurafenib-treated melanoma patients. It is anticipated that this mutation would confer comparable resistance to the MAPK pathway, similar to COP1 mutations, as DET1^S192F^ also fails to degrade ETV5 when combined with COP1^WT^ in 293T cells (Figure 2H). Recently published DET1 structures [13,14] suggest that this variant does not disrupt interactions with its interaction partners (COP1, DDB1, DDA, UBE2E family enzymes). However, we suspect that the missense mutation impacts DET1 protein stability because our efforts to reconstitute *DET1^KO^* A375 cells to express DET1^S192F^ at levels equal to WT protein were unsuccessful. The mutant protein consistently displayed lower expression. Further investigation is needed to elucidate how DET1^S192F^ impairs CRL4^COP1/DET1^ function.

## Figures and Tables

**Figure 1 cells-14-00975-f001:**
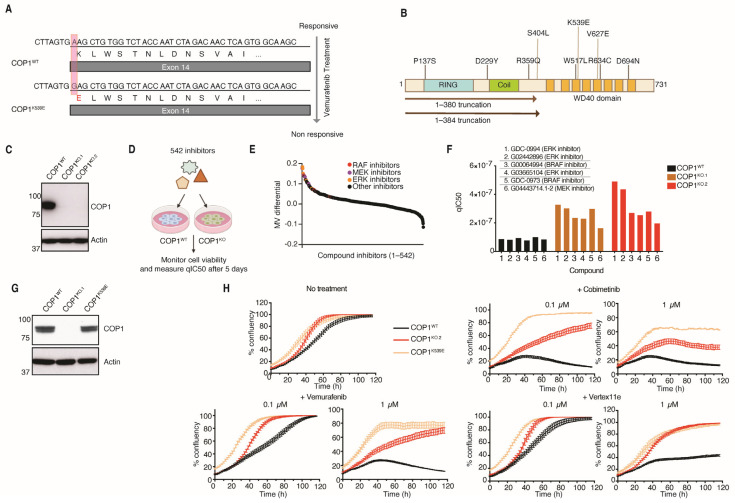
Deletion or mutational inactivation of COP1 mediates resistance to MAPK pathway inhibitors. (**A**) DNA sequence of the *COP1^K539E^* missense mutation. (**B**) Domain organization of the COP1 protein indicating the mutations found in melanoma patients treated with Vemurafenib. (**C**) Western blots of A375 cells (*COP1^WT^*) and two COP1-deficient clones generated using CRISPR/Cas9 technology (*COP1^KO.1^* and *COP1^KO.2^*). (**D**) Chemical genomic screen strategy using *COP1^WT^* and *COP1^KO^* A375 cells. (**E**) The graph indicates the mean viability (MV) differential (y axis) between *COP1^WT^* and *COP1^KO^* A375 cells treated with the compounds in (d; x axis). (**F**) The graph shows the qIC50 for different MAPK pathway inhibitors in A375 cells. (**G**) Western blots of A375 cells. (**H**) The graphs indicate A375 cell proliferation. The results are representative of 3 independent experiments.

**Figure 2 cells-14-00975-f002:**
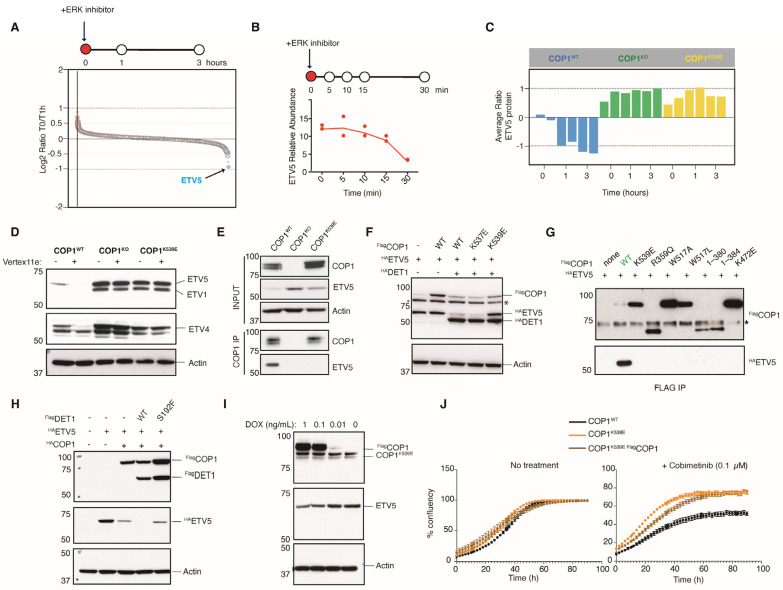
COP1^K539E^ fails to promote ETV5 degradation upon ERK inhibition. (**A**) Global proteomic analysis of A375 cells before and after a 1 h treatment with 1 µM ERK inhibitor Vertex11e. The graph indicates the change in abundance (y axis) for detected peptides (x axis). The circles indicating peptides unique to ETV5 are colored blue. (**B**,**C**) The graphs show the change in ETV5 protein abundance in A375 cells after treatment with 1 µM Vertex11e. The circles in (**B**) indicate the results from individual biological replicates (*n* = 2). The bars in (**C**) represent the mean of two biological replicates. (**D**) Western blots of A375 cells treated with 1 µM Vertex11e for 1 h. (**E**) COP1 is immunoprecipitated (IP) from lysates of WT or *COP1^K539E^* A375 cells using an anti-COP1 antibody (28A4, Genentech) and magnetic protein A/G beads. The input lysates and immunoprecipitates were subsequently blotted against ETV5. *COP1^KO^* A375 cells were used. (**F**–**H**) Western blots of transfected 293T cells. WT and mutant genes are overexpressed using a PRK5-based mammalian expression, introduced into 293T cells via Lipofectamine transfection reagent. (**I**) Western blots of *COP1^K539E^* A375 cells expressing doxycycline-inducible ^Flag^COP1. Cells were treated with doxycycline for 24 h. (**J**) The graphs indicate A375 cell proliferation. The results are representative of 2 independent experiments. * Non-specific band.

**Figure 3 cells-14-00975-f003:**
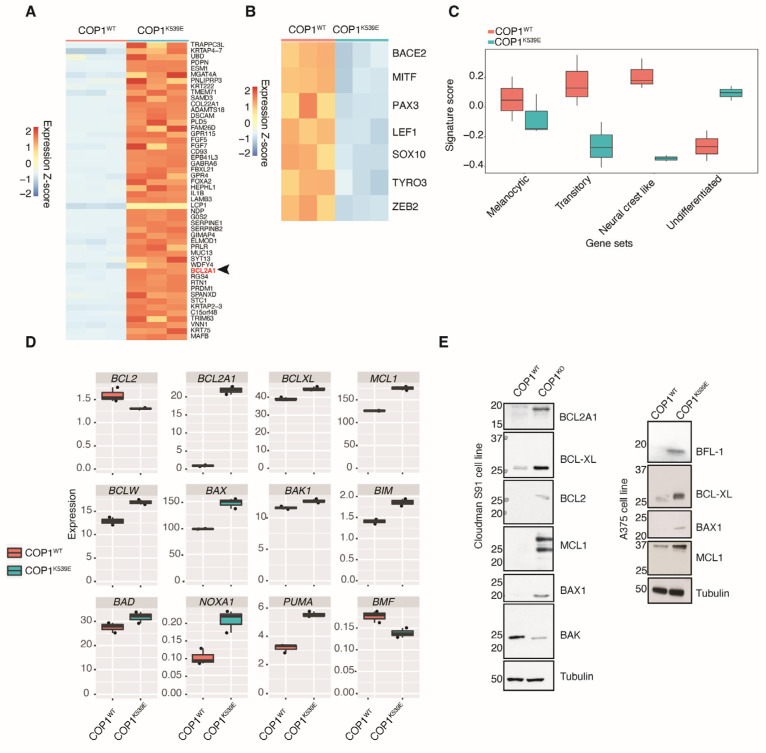
*COP1^K539E^* A375 melanoma cells exhibit altered expression of genes regulating cell survival and differentiation. (**A**,**B**) Heatmaps show gene expression for the top 50 differentially expressed genes between *COP1^WT^* and *COP1^K539E^* A375 cells (**A**) and differentially expressed genes involved in melanocyte differentiation (**B**). (**C**) Boxplots of the gene signature score using the melanoma subtypes defined in Tsoi et al. [36] across WT and COP1 SNP genotypes. (**D**) Box and whisker plots of gene expression in A375 cells. Black circles indicate 3 biological replicates. (**E**) Western blots of A375 and Cloudman S91 cells.

**Figure 4 cells-14-00975-f004:**
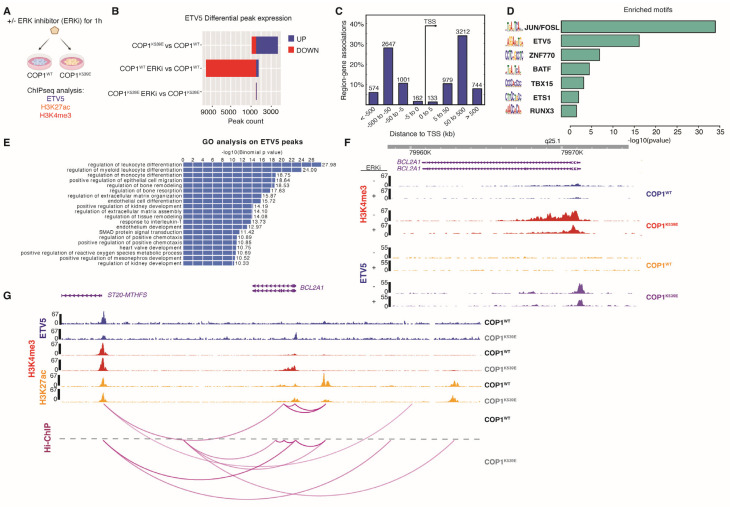
Analysis of the chromatin in *COP1^K539E^* A375 cells. (**A**) Scheme for ChIP-seq analysis of *COP1^WT^* and *COP1^K539E^* A375 cells. Cells were treated with 1 µM Vertex11e. (**B**) The bars indicate the number of ETV5 binding sites showing differential occupancy in the A375 cells indicated. (**C**) The histogram shows the distribution of ETV5 peaks relative to transcription start sites (TSS) in *COP1^K539E^* A375 cells. (**D**) The graph indicates transcription factor motifs associated with ETV5 distal binding sites in *COP1^K539E^* A375 cells. The *p*-values are determined by the hypergeometric test. (**E**) Gene ontology (GO) terms associated with ETV5 distal binding sites in *COP1^K539E^* A375 cells. The *p*-values are determined by the hypergeometric test. (**F**) Traces indicate normalized H3K4me3 ChIP-seq reads at the *BCL2A1* promoter in A375 cells. The location of the *BCL2A1* coding region is shown in purple at the top. (**G**) Traces show normalized ETV5 (blue), H3K4me3 (red), and H3K27ac (orange) ChIP-seq reads around the *BCL2A1* gene. Hi-ChIP arcs (magenta).

**Figure 5 cells-14-00975-f005:**
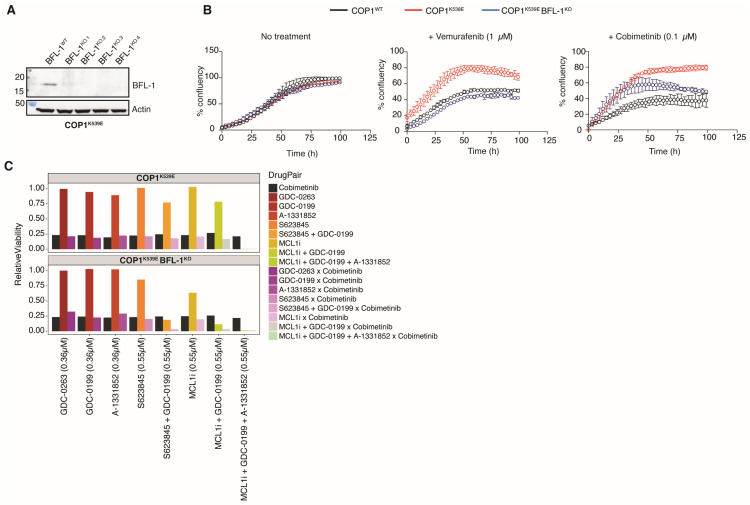
Elevated *BCL2A1* expression in *COP1* mutant cells drives resistance to MAPK pathway inhibitors. (**A**) Western blots of A375 cells. (**B**) Graphs indicate A375 cell proliferation. Results are representative of 2 independent experiments. (**C**) Relative viability of A375 cells upon treatment with different inhibitors of the BCL2 pro-survival proteins, GDC-0199 targeting BCL2, A-1331852 targeting BCL-XL, and AMG-176 and S63845 targeting MCL1, alone or in combination with the MEK inhibitors cobimetinib (GDC-0973). The black line shows the response to cobimetinib at 0.18 μM, yellow-red shades show the response to the BCL2-family inhibitor at respective concentrations (see x-axis), and purple-blue shades show the response to the corresponding combination of drugs.

## Data Availability

All data is available upon request from Ada Ndoja (ndojaa@gene.com). The RNA-seq data have been deposited in GEO with the identifier GSE203470, and the proteomics data have been deposited in MassIVE with the accession MSV000097835 (login MSV000097835_reviewer and password COP1BRAF). Unique reagents generated in this study may be requested through Genentech’s MTA program.

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
