# Peer review of "COP1 Deficiency in BRAF^V600E^ Melanomas Confers Resistance to Inhibitors of the MAPK Pathway"

_cells, 2025, doi:10.3390/cells14130975_

Round 1
Reviewer 1 Report
Comments and Suggestions for Authors
This is a well-written and well-executed study that uncovers how secondary loss-of-function (LOF) mutations in COP1confer resistance to MAPK pathway inhibitors in melanoma. The genetic work is carefully performed, and the validation of changes in protein expression is sufficiently robust. A promising direction for future research would be to investigate how targeting COP1-mediated resistance—potentially via BFL-1 inhibitor combinations—could help overcome therapeutic failure. A few clarifications and suggestions could further strengthen the manuscript. Overall, the work is close to publication quality.
“Two COP1-deficient clones generated using CRISPR/Cas9 technology (COP1KO.1 and COP1KO.2)”
- What distinguishes COP1KO.1 from COP1KO.2? Are they identical in the targeted region, or do they differ in the precise deletions/mutations?
- Were any flanking regions near COP1 disrupted during the editing process?
- Including a schematic similar to Figure 1A would help clarify the exact edits in each clone and confirm specificity.
“Graph indicates the mean viability (MV) differential (y-axis) between COP1WT and COP1KO A375 cells treated with the compounds in (d; x-axis).”
- Out of curiosity, some inhibitors show MV < 0, meaning the COP1KO cells are more sensitive than WT. What are these compounds?
- This raises an interesting possibility: LOF of COP1 may sensitize cells to other drugs, which could be leveraged therapeutically to combat MAPK-inhibitor resistance.
- A short discussion of this observation could open useful avenues for future work.
In Figure 1H,
- It appears that COP1KO cells respond more strongly to Vemurafenib and Cobimetinib than the COP1K539E mutant.
- Could there be off-target effects in the CRISPR-generated COP1KO cells that increase drug sensitivity or reduce viability?
- Some discussion or additional validation (e.g., rescue experiments) would help clarify this.
Do COP1K539E and DET1S192F mutations confer the same level of resistance to MAPK pathway inhibitors? Do they influence ETV5 expression to the same extent, or are their downstream effects distinct?
Reviewer 2 Report
Comments and Suggestions for Authors
The manuscript entitled " COP1 deficiency in BRAFV600E melanomas confers resistance to inhibitors of the MAPK pathway " by Ada Ndoja and coworkers examined the role of CRL4 Cop1/DET1 in acquired resistance to MAPK inhibitors in BRAF mutated melanoma cells. The research is interesting, well organized and detailed.
Here are the comments:
- In Introduction section COP1/DET1 complex is not sufficiently explained.
- In 1. Cell culture section lack the information on genetic background of the cells used.
- The data on antibody dilution would be helpful as well more explanation of the experimental procedure, e.g. western blot detection, knock down of ETV5 and COP1 etc.
- How the COP1 mutations were determined?
- Please, explain the value of qIC50.
- Figure 1E, it is not clear what the authors show here.
- Figure 2E, IP must be explained in figure legend.
- HEK293T cells have to be written in a unique way, and should be mentioned in Materials and methods section.
- A graphical abstract of a scheme showing the idea of the research should be a benefit.
Reviewer 3 Report
Comments and Suggestions for Authors
As the effect of COP1/DET1 loss on BRAFi resistance was previously published by another group in 2018 (their ref 27), the question is whether the current study has substantially added new data. The answer is yes, particularly the characterization of a new missense COP1 mutation K539E from a patient sample that was knocked into a cell line, and additional new overexpressed mutants that were found to not associate with its normal substrate ETV5 in a dominant negative fashion over WT COP1. They also knocked out ETV5 and found that it resensitized COP1-KO cells. In contrast to the previous study, COP1 loss did not affect DUSP4/6, but rather induced an EMT-like state known to be associated with MAPKi resistance and operates at least partially through BCL2A1 overexpression. Overall, the work is well conducted, but several concerns must be addressed below.
- The Introduction should more clearly describe the relative roles of COP1 and DET1 in CRL4 activity and how they are related to one another.
- Section 3.1, it should be clearly stated which set of exome data were examined to identify patient samples with COP1 mutations. Are these their own data? Publicly available? How is the response to BRAFi known? I can’t find any information in the Methods either.
- Section 3.1, “modifying the endogenous COP1 locus to express COP1-K539E” this should be more clearly described how this was done.
- Section 3.3, it should be clearly stated how the various mutants were expressed. Are these lentiviral?
- Authors should be more up-front about ref 27 having already made many of the same observations, right now mentions of it are very subtle. The authors should state very clearly what new biological discoveries they have made on top of ref 27. Relevantly, the Conclusions is too short and the perfect place to put these. Disclaimer: I am not associated with ref 27 in any way.
- Section 3.5, “To confirm that ETV5 deficiency allowed the COP1KO 389 ETV5KO cells to proliferate in the presence of cobimetinib” I think this is a typo? ETV5 deficiency caused the cells to NOT proliferate +MEKi.
- Does COP1-KO also cause EMT and BCL2A1 changes similarly to COP1-K539E? What about ETV5 KO affects on these genes?
- The triple and quadruple drug combinations in Fig. 5 are likely just toxic and it is not clear whether they add anything to the story. If authors insist on keeping them, they should also be tested in normal melanocytes to see if they are just generally toxic or not. Also, CDK9 is mentioned in Fig. 5C, but not anywhere in the manuscript. Finally, fig. 5C legend needs more description, it is rather confusing and not clear what exactly what is being shown.
Round 2
Reviewer 3 Report
Comments and Suggestions for Authors
Authors have addressed most of my comments, but I still find Fig. 5C problematic. I think it would just be best to remove CDK9 inhibitor data as it really adds nothing to the study at all. As I initially said, the 4+ drug combinations are also uninformative, as most likely they would kill any cells, not just melanoma. Those groups of bars (#5 and 6 from the left) should just be removed. Finally, the legend is still very difficult to follow. It's very hard to match up the colors to the legend and they are presented in different orders. It is very difficult to draw a conclusion from the graph.
Author Response
We appreciate the reviewer's careful assessment of our manuscript. We agree with the feedback and have revised Figure 5C by removing the data involving the CDK9 inhibitor. The legend and labels have also been updated for clarity. We believe these changes significantly improve the interpretability of the data and conclusions. The revised manuscript is uploaded.
